# Examining the burden and relationship between stunting and wasting among Timor-Leste under five rural children

Paluku Bahwere[1]*, Debra S. Judge[2], Phoebe Spencer[2], Faraja Chiwile[3], Mueni Mutunga[3]

1 ActionAgainst Hunger UK, London, United Kingdom, 2 School of Human Sciences, University of Western Australia, Perth, Australia, 3 United Nations Children's Fund (UNICEF) East Asia Pacific Regional Office, Bangkok, Thailand

☯ These authors contributed equally to this work.
* paluku.bahwere@gmail.com

**Data Availability Statement:** The datasets used to generate the results presented in the paper can be accessed at the UWA Research repository using the link https://doi.org/10.26182/zvkz-e569.

## Abstract

Globally and in Timor-Leste, wasting and stunting remain major public health problems among 'under five years children, but the interrelationship between the two has been poorly investigated. A better understanding of this interrelationship is a prerequisite to improving wasting and stunting programming. In our study, we assessed the influence of age on the prevalence of wasting and stunting, the overlap between the two conditions, and the effect of wasting parameters on linear growth catch-up using the data of 401 children recruited at 0 to 54 months of age [median (IQR) of 17 (7–32) months] with repeated anthropometric assessments [median (IQR) follow-up time was 25 (16–39) months]. At recruitment, prevalences of stunting, wasting and concurrence of the two conditions were 54.6%, 9.5% and 4.6%, respectively. These prevalences were already high and above the thresholds for public health importance among children below months of age and remained high throughouttheir childhood. Over the follow-up period, the change (95%CI) in Height-for-Age Z-score (HAZ) was -0.01 (-0.13; 0.11) (p = 0.850), and that of the Height-for-Age Difference (HAD) was -3.74 (-4.28; -3.21) cm (p<0.001). Stunting reversal was observed in 25.6% of those stunted at recruitment, while a positive change in HAD was observed in only 19.6% of assessed children. Path analysis by structural equation modelling showed no significant direct effect of WHZ at recruitment on the likelihood of positive change in HAD, with its influence being fully mediated by its change over the follow-up period. This change had an inverse relationship with the occurrence of a positive change in HAD. On the contrary, Mid-Upper Arm Circumference at recruitment had a significant positive direct effect on the likelihood of a positive HAD change. These results show that interventions to combat wasting and stunting need to be integrated.

**Funding:** The original study was funded by the Australian Research Council (Grant DP 120101588), the School of Anatomy, Physiology, and Human Biology at the University of Western Australia, and the Timor-Leste UNICEF office. This particular study (secondary data analysis and paper writing) was funded by the European Union through the UNICEF Timor-Lester office. The funders played no role in the design of this secondary data analysis and the interpretation of the findings.

**Competing interests:** All authors have no competing interests.

## Introduction

Undernutrition is a major global public health problem, especially in low- and middle-income countries in Africa and Asia [1–6]. In 2022, 45 million and 148 million children under the age of five years worldwide suffered from wasting and stunting, respectively [7]. In all affected countries, undernutrition leads to serious public health risks, hinders development, and causes huge economic loss [4, 8–10]. It is believed that global and country-level improvements in nutrition will significantly accelerate the achievement of many of the Sustainable Development Goals (SDGs) by contributing to ensuring healthy living of the populations, reducing poverty, and achieving education and employment goals [11, 12].

In Timor-Leste, the prevalence of child undernutrition remains high, with stunting at 47.1%) and wasting at 8.6%, classifying the country among those in which undernutrition is a major public health problem [13, 14]. Household food insecurity in Timor-Leste is high. National estimates from the 2020 Food and Nutrition surveys that used the Food Insecurity Experience Scale (FIES) indicate that 50.4% of Timorese households were food insecure in 2020, with this percentage being 55% for rural residents [13]. Infants and young children whose typical complementary food is watery rice porridge have year-round insufficient intake of important growth-promoting nutrients [15–17]. The seasonal variation of intake also exposes these children to recurrent episodes of wasting process as measured by low weight for age and low weight relative to length/height, potentially leading to the accumulation of linear growth deficit over time [15, 18, 19]. It is likely that in Timor-Leste, as in most affected countries, children from rural areas are more affected by these factors, leading to a higher prevalence of all forms of undernutrition, including vitamin and mineral deficiencies, given their overreliance on autosubsistence farming [20–23].

Reducing the level of all forms of undernutrition is one of the strategies of the Government of Timor-Leste for reaching its Sustainable Development Goals by 2030. In this regard, the Government of Timor-Leste, through the multispectral nutrition programming approach, has put in place policies and strategies enabling the implementation of several nutrition specific and nutrition sensitive interventions aiming at addressing all forms of undernutrition specifically targeting pregnant and lactating women, children aged less than five years and adolescent girls [24, 25]. Prioritized nutrition specific interventions include interventions in the areas of infant and young children feeding (promotion of exclusive breastfeeding, promotion of timely and appropriate complementary feeding practices), micronutrient supplementation or fortification (vitamin A supplementation, iron-folic acid supplementation for pregnant and lactating women and adolescent girls, deworming program for children 1 to 5 years old, multiple micronutrients supplementation for children 6 to 23 months old, cases treatment of vitamin A deficiency), dietary supplementation (maternal dietary supplementation, dietary supplementation for children), treatment of severe acute malnutrition, and diseases prevention and treatment [24, 25]. The currently prioritized nutrition sensitive interventions include agriculture and food security, social safety nets, early child development, maternal mental health, women's empowerment, child protection, classroom education, health and family planning, and water and sanitation [24, 25]. Though the list includes interventions dedicated to addressing wasting and those dedicated to addressing stunting, similar to what has been observed in many affected countries, the differences in targeting criteria and mechanisms due to differences in defined focus and objectives and the implementation in silos by different and uncoordinated national and international organizations continue to replicate the separation between wasting and stunting programing ignoring the interrelation between the two [23, 26, 27]. In recent years, there has been a renewed interest in examining the relationship between wasting and stunting to find ways to improve strategies to combat these two forms of child undernutrition [28–31].

Historically, wasting and stunting have been considered as distinct forms of undernutrition and addressed with different strategies and programs. This was due to the limited understanding of the relationship between the two conditions. At a global level, rapidly growing evidence of sharing of determinants, concurrence of the two forms and potential linkage of the processes leading to their occurrence is closing this knowledge gap [29, 30, 32–36].

The relationship between wasting and stunting may be context-specific [37, 38]. Examining this relationship is a prerequisite to developing policies and interventions that can address these conditions effectively in Timor Leste. Despite the increased knowledge on the determinants of the nutritional situation of Timorese children described above since the country's independence, the interrelationship between wasting and stunting has not yet been examined in depth. This study aims to strengthen that evidence by determining the relationship between the prevalence of wasting and stunting and age in children under the age of five years, describing the overlap between these two forms of undernutrition, and identifying the effects of wasting parameters on linear growth catch up.

## Methods

### Study design

This is a secondary analysis of the sub-set of data of a study that measures changes at the population level over time using a repeated cross-sectional survey design [15, 18, 39–41], also called pseudo-longitudinal by some statisticians [42, 43]. The dataset was shared with us by the University of Western Australia (UWA) research team that conducted the original surveys. The design of this original survey involved multiple cross-sectional surveys with resampling of households of the targeted sub-districts at each round [18, 39]. This secondary analysis used data for children who were sampled at least twice at an age below 5 years. These data were extracted from the full database of the original study by its principal investigator, who made a copy of. the de-identified database to us in Excel format in July 2022. All the surveys that had been conducted by then contributed study subjects to our study, and no variable was excluded. According to the study objective, selected children were analyzed using a cross-sectional approach (pooling data from the different cross-sectional analyses) and a longitudinal analysis approach.

### Setting

Data used in this study were collected in Timor-Leste, a country of approximately 1.3 million people, 70% of whom live in rural areas. The country is administratively divided into 12 municipalities or districtsand one special administrative region, 65 administrative posts or subdistricts, 452 sucos (villages), and 2,233 aldeias (hamlets). Approximately 80% of the economically active population has subsistence agriculture as the main source of income. The Governement of Timor-Leste provides health services to its population through a network of hospitals, community health centers, health posts, and integrated outreach services. The number of these facilities has been increasing since the country's independence in 2002. In 2018, there were one tertiary hospital, five referral regional hospitals, 71 community health centers, 323 health posts, and 459 integrated outreach services sites. All these provide preventable and/or curative nutrition services.

Data for this study were collected in central areas (near market and clinic) and more peripheral areas of Ossu (central mountain region, Viqueque municipality) and Natarbora (south coastal plains, Manatuto municipality) sub-districts of Timor-Leste over twelve rounds of surveys run from 2009 to 2020. Some of the data collection occurred during the harvest season (June-November) and some during the lean season (March-May).

## Data collection and procedures

Field data collection methods have been extensively described elsewhere [40, 44]. In summary, administrative, child and household-related information was collected in each round by a multidisciplinary research team consisting of UWA researchers and specially recruited and trained local research assistants. The team interviewed one adult from each of the sampled households to collect administrative and socio-demographic information as well as information on household resources and used standard anthropometric measurement methods to determine weight, length/height, and mid-upper arm circumference [40, 45] to measure each child resident in the household. Administrative information included the month of data collection, household identifiers, and child identifier. Child-related data collected included socio-demographic information (age and sex) and anthropometric nutrition parameters (Height/length, weight, and mid-upper arm circumference). Household related information collected included head of the households' characteristics (sex, education level), mother characteristics (education level), siblings (number), household size (total number of people, number younger than 18 years, number aged less than 5 years) and water and sanitation characteristics (source of drinking water type, toilet type).

Administrative, child socio-demographic data and household related information were collected via interviews with the female head of the household or directly measured by the enumerator. Anthropometric measurements were performed according to standard technique [46].

## Data management and variables specification

The data was shared electronically in the form of Microsoft Excel files. These files were converted into STATA 14.0 format and merged into wide format. We used the wide format for data cleaning and new variable creation. Finally, the wide format was converted into a long format. We used the wide or the long format as appropriate during data analysis.

The new variables created included the binary variables for describing the nutrition status derived from the continuous weight-for-height Z-score (WHZ), height-for-age Z-score (HAZ), and Mid-Upper Arm Circumference (MUAC) using the standard WHO cut-off for the definition of wasting and stunting [47, 48], variables representing the change in nutrition status calculated as attained value at last assessment minus the value at recruitment, the variable season of data recruitment derived from the date of data collection, the duration into the study in months obtained by calculating the number of months that elapsed between recruitment and last assessment using the relevant dates and height-for-age difference variables fully described in Table 1 below.

## Study principal endpoints

The study's principal endpoints were 1. prevalence of stunting among enrolled children, 2. prevalence of wasting among enrolled children, 3. correlation between wasting prevalence and age and between stunting prevalence and age, 4. change in linear growth parameter during follow up, and 5 effect of wasting parameters at recruitment on linear growth catch-up.

## Statistical analysis

All the statistical analyses were performed using the statistical software STATA version 14.0 (StataCorp, College Station). We used means and their standard deviations (SD), medians and their interquartile range (IQR) and percentages and their 95% confidence interval (95%CI) to summarize the continuous and categorical variables as appropriate. Multilevel linear mixed

**Table 1. Created variables and specifications.**

| Variable | Specification/Definition |
|---|---|
| Height-for-Age Difference (HAD) from expected | New nutrition indices were proposed by Leroy et al. [48].<br>HAD is calculated by comparing a child's height/length to the average height/length of healthy children of the same age and sex. The formula is:<br>Observed height/length (cm) minus the age and sex-specific median of height/length (cm) |
| Height-for-Age Difference change | Describes the evolution of the new nutritional indice HAD over time. It is calculated as follows: Final Height-for-Age Difference (cm) minus recruitment Height-for-Age Difference (cm) $HAD_f$—$HAD_i$ (cm) |
| Height-for-Age Difference change category | Binary variable of Height-for-Age Difference change in cm (0 = no change or decrease; 1 = increase) |
| Wasted | Binary variable of weight-for-height/length (WHZ) based on 2006 World Health Organization growth standards<br>(0 = WHZ$\geq$-2; 1 = WHZ$<$-2) |
| Stunted | Binary variable of height/length-for-age (HAZ) based on 2006 World Health Organization growth standards<br>(0 = HAZ$\geq$-2; 1 = HAZ$<$-2) |

effects models were used to examine the association between the dependent and independent variables. Linear regression modelling is used throughout regardless of the forms of the curves of actual data, as the study objectives were only to know whether there is an association between the dependent and the independent variables and whether the increase in the considered independent variables cause an increase in the dependent variable [49].

Mediation analysis performed by the Structural Equation Modeling (SEM) statistical method (STATA command sem) was used to determine the total, direct, and indirect effects of WHZ and MUAC on the HAD gain [50]. The SEM can use several regression models, but in this study, we used the linear regression models to calculate the direct and indirect coefficients for the different paths forming our hypothesized models [50]. The indirect effect of the different explanatory variables were calculated using the product method [50]. In summary, the linear regression equations tested included change in MUAC and WHZ as potential mediators of the model while age at recruitment and at last assessment, HAD at recruitment, and elapsed time between recruitment and last assessment were included as covariates. The inclusion of WHZ and MUAC as potential mediators is because they are currently widely used in surveys and programs to detect wasting in children 6 to 59 months of age and in the monitoring of the response to nutrition rehabilitation interventions [51–55]. In addition, several studies have demonstrated that variation in WHZ predicts subsequent linear catch up growth that occurs with a time lag of three months or more after the episode of accelerated weight gain [29, 30, 56, 57].

Similarly, data analysis from a longitudinal study conducted in Cambodia in children under 5 years old revealed that MUAC change also had a direct correlation with linear growth, with the change in MUAC and height occurring concurrently (29). A probability value (p) less than 0.05 was deemed significant. Scatterplots were used to depict the overlap between two continuous variable variables.

## Ethics considerations

No specific request was sought, and no special ethical approval was applied for this secondary data analysis as this exercise is covered by the approvals granted to the original study. The original study received ethical approval from the Human Research Ethics Committee of the University of Western Australia (2019/RA/4/1/2401), the Timor-Leste Ministry of Health (2009–

2010 rounds) and the Timor-Leste Cabinet of Health Research and Development (MS-CHRD/ IV/2011/23), the Timor-Leste Gabinete Pesquisa no Dezenvolvimento Saude (GPDS/MdS/IX/ 2012, covered data collection rounds 2012–2015), and the Timor-Leste Ministerio da Saude Instituto Nacional da Saude (MS-INS/GDE/DP-EA/V/2016, MS-INS/GDE/DP-EA/IV/2017/ 528, 103MS-INS/DE-DP/CDC-DEP/II/2018/IV/2017/528, 182MS-INS/DE/II/2020). Participation was on a voluntary basis. Informed verbal consent, witnessed by the community leaders, was obtained from the children's parents or guardians before enrollment in the study. The process of obtaining the consent to participate has been previously described [18, 39–41]. We used a de-identified database; hence, the anonymity of all the participants was preserved throughout the data analysis process.

## Results

### Participants description

Data of 401 children aged less than five years enrolled between 2009 and 2018 (Fig 1) and measured at least twice were available for our analysis, of whom 198 (49.4%) were from Ossu and 203 (50.6%) from Natarbora. These children were from 195 households. The sample comprised 205 boys (51.1%) and 196 girls (48.9%). The age at enrolment ranged from 0 to 54 months with a median (IQR) of 17 (7–32) months. The study participants had one to seven repeated measurements. The cross-sectional analysis uses all available measurements, while the mediation analysis uses the first and last measurements.

### Cross-sectional analysis of the pooled data

Data pooling of all the successive cross-sectional surveys gave a total of 1409 measurements for MUAC, 1410 for weight and WAZ, 1369 for height and related indices (HAZ and HAD), and 1368 for WHZ.

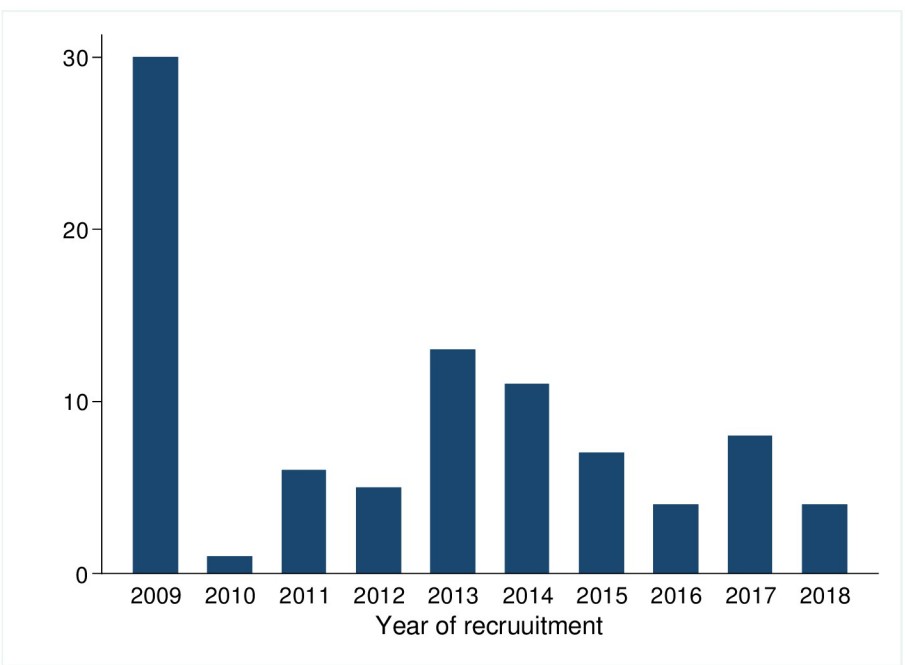

**Fig 1. Number of children recruited in the different years.**

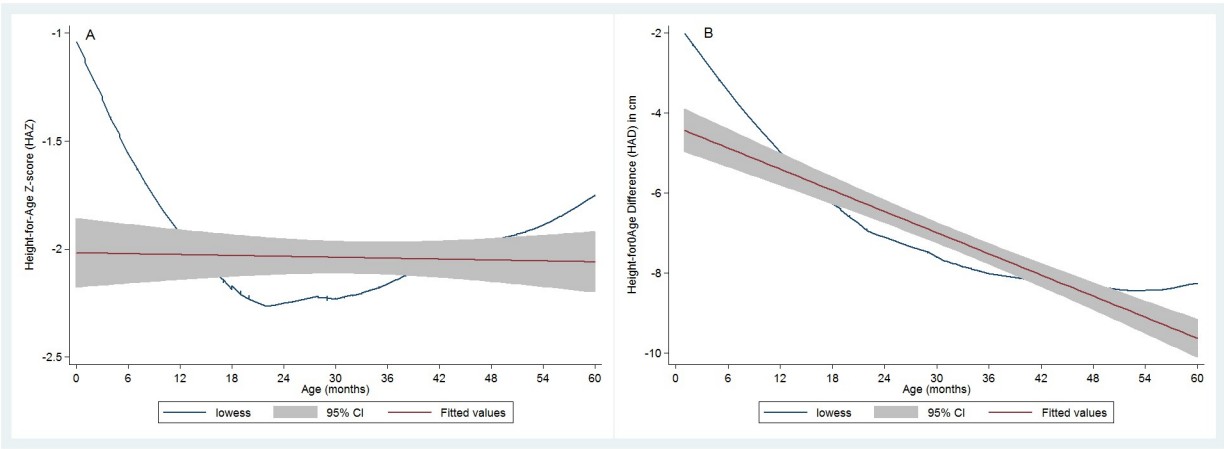

**Fig 2. Relationship between age and length/height-for-age Z-score and between length/height-for-age difference and age.**

**Stunting.** The pooled prevalence (95% CI) of stunting was 54.6% (52.0; 57.3) by HAZ<-2 Z-score criterion; the prevalence of children with any level of height deficit (HAD<0 cm) was 93.0 (91.6; 94.3) %. Fig 2 below depicts the relationship between linear growth status indicators and age. As evidenced by the LOWESS curves, there was a non-linear relationship between the continuous variable age and HAZ and HAD. The correlation was not significant for HAZ [$r_s$ = 0.032; p = 0.233] but significant for HAD [$r_s$ = -0.317; p<0.001]. The LOWESS curve had a U form for the HAZ and age relationship, suggesting some increase in linear growth in children above 24 months of age (Fig 2A), which translated into a stabilization of height deficit from around 42 months of age (Fig 2B). The comparison of the HAZ and HAD LOWESS curves revealed a discrepancy of pattern with the HAZ curve suggesting catch up linear growth after 24 months, while the HAD curve indicated a widening of the height deficit until 42 months with almost no reduction of the deficit until the age of 60 months (Fig 2A and 2B).

Table 2 below describes the prevalence of stunting and height deficit according to the defined age groups. As shown, there was a non-linear significant association between age in groups and stunting prevalence, with this prevalence being the lowest among infants in the 0 to 5 months age group and the highest among those in the 12 to 23 months age group. This prevalence was already high among the youngest age group, and its reduction among children older than 23 months was of limited magnitude (Table 2). The proportion of children whose height was below the sex and age-specific reference median height was also well above the expected 50% of the youngest age group, with this proportion increasing linearly with age (Table 2).

Children measured in Ossu surveys were shorter and had a significantly lower mean of HAZ than those measured in Natarbora surveys [mean (SD) = -2.5 (1.3) for Ossu children versus -1.7 (1.3) for Natarbora children; Δ (95%CI) = -0.8 (-1.0; -0.7); p<0.001] and a significantly lower HAD [mean (SD) = —8.9 (4.7) cm for Ossu children versus -5.9 (4.5) cm for Natarbora children; Δ (95%CI) = -3.0 (-1.0; -0.7) cm; p<0.001]. The HAZ and the HAD were not influenced by either the sex of the child or the season of the survey in the analyzed sample.

**Wasting.** The overall mean (SD) of WHZ and MUAC were -0.5 (1.3) and 147.2 (11.9) mm, respectively. Children measured in Ossu surveys had a higher mean of WHZ than those measured in Natarbora surveys [mean(SD) = -0.4 (1.2) for Ossu children versus -0.6 (1.3) for Natarbora children; Δ (95%CI) = 0.2 (0.1; 0.4;p = 0.001] but that of MUAC did not differ [mean(SD) = 147.3 (11.7) for Ossu children versus 147.2 (12.1) for Natarbora children; Δ (95%

**Table 2. Association of stunting and height deficit prevalences with age.**

| Age category | n | Unadjusted analysis[1] | | | adjusted analysis[1] | | |
|---|---|---|---|---|---|---|---|
| | | %[2] | OR[3] | p-value | %[2] | AOR | p-value |
| *Height-for-age Z-score (n = 4264)* | | | | | | | |
| 0–5 months | 88 | 30.7 | 1.0 | | 33.7 | 1.0 | |
| 6–11 months | 108 | 41.7 | 1.6 (0.9–2.9) | 0.114 | 45.2 | 3.2 (1.2–8.1) | 0.013 |
| 12–23 months | 237 | 65.8 | 4.3 (2.6–7.4) | <0.001 | 67.5 | 14.6 (6.3–33.8) | <0.001 |
| 24–47 months | 611 | 59.4 | 3.3 (2.0–5.3) | <0.001 | 60.6 | 10.6 (4.8–23.3) | <0.001 |
| 48–60 months | 325 | 48.3 | 2.1 (1.3–3.5) | 0.004 | 48.3 | 3.9 (1.7–9.0) | 0.001 |
| *Height-for-age Difference)* | | | | | | | |
| **0–5 months** | 88 | 79.5 | 1.0 | | 80.0 | 1.0 | |
| 6–11 months | 108 | 86.1 | 1.6 (0.7–3.4) | 0.224 | 86.6 | 3.8 (0.9–16.1) | 0.070 |
| 12–23 months | 237 | 93.2 | 3.6 (1.7–7.3) | 0.001 | 93.5 | 13.0 (3.3–50.5) | <0.001 |
| **24–47 months** | 611 | 95.1 | 5.0 (2.6–9.4) | <0.001 | 95.3 | 38.3 (8.9–165.6)- | <0.001 |
| 48–60 months | 325 | 94.8 | 4.6 (2.3–9.5) | <0.001 | 94.8 | 36.1 (6.5–199.5) | <0.001 |

[1]Computed using multilevel regression modeling,

[2]% = prevalence;

[3]OR = Odds ratio and 95% confidence interval;

[4]AOR = Adjusted odds ratio and 95% confidence interval; not retained variables = sex and season of data collection

CI) = -0.1(-1.4; 1.1;p = 0.858]. There was a significant non-linear relationship between WHZ or MUAC and the continuous variable age, with the Spearman correlation coefficient being-0.255 (p<0.001) and 0.383 (p<0.001) for WHZ and MUAC, respectively. With regards to prevalence, wasting (WHZ<-2 criterion) was observed in 9.5 (7.9; 11.0) % of measured children, while only 1.9 (1.0; 2.8) % of them were wasted by MUAC<125 mm criterion. There was no association between wasting prevalence by WHZ criterion and the age group (p = 0.510), and the change across the age group was non-linear (p = 0.546), with the lowest prevalence encountered in the <6 months age group and the highest among the 12–23 months age group (Fig 3A). In contrast, there was a significant association between wasting prevalence MUAC criterion (MUAC<125 mm) and age group (p<0.001), with this prevalence decreasing linearly

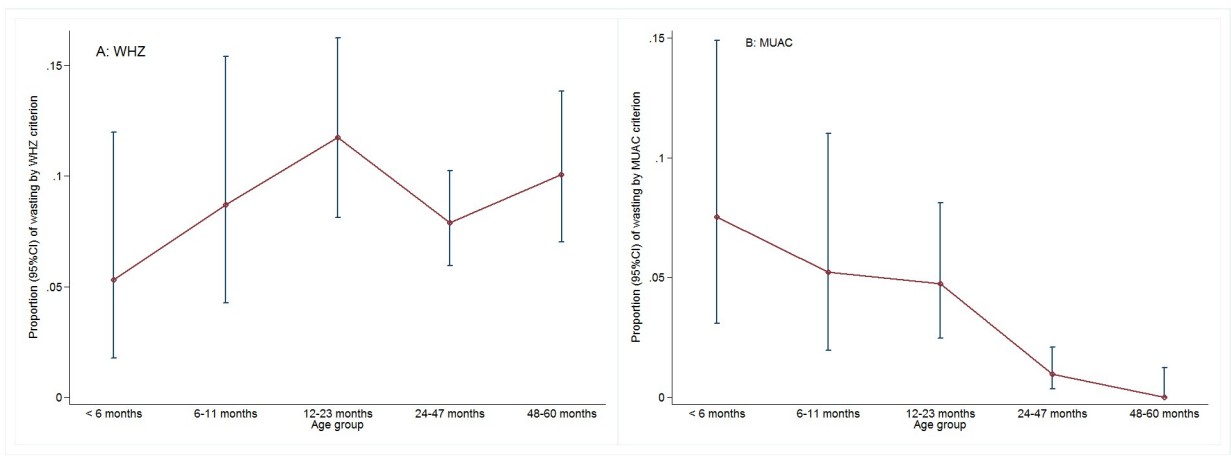

**Fig 3. Weight-for height Z-score and mid-upper arm circumference based-wasting prevalence across the age groups for children included in the pooled analysis.**

(p<0.001) from the younger age group to the group with oldest children of the age range (Fig 3B). Sex and season had no influence on the wasting prevalence as defined by WHZ, but there were significantly more cases of wasting by MUAC criterion when the data were collected during the lean season [OR (95%CI) = 3.0 (1.3–7.1); p = 0.012]. The strength of the association remained almost the same after adjustment for the sex and the age of the child and taking into account a possible clustering at the subdistrict level [OR (95%CI) = 3.0 (1.2–7.7); p = 0.019].

**Stunting and wasting overlap.** Concurrence of wasting and stunting was found in 4.6% (63/1368) measurements, with the prevalence (95%CI) adjusted for clustering at individual and subdistrict levels being 4.3 (2.9–5.8) %. The age mean of those concurrently wasted and stunted was 35.2 (31.5–38.9) months, and boys represented 57.1% of the group. The distribution across seasons of their measurement was 50.8% (32/63) in the lean season and 49.2% (31/63) in the post-harvest season. Their average (95%CI) MUAC was 135 (133–137) mm. Almost half of those found to be wasted were also stunted (63/130 = 48.5%). The likelihood of being wasted and stunted was not influenced by child age [AOR = 1.02 (1.0–1.04); p = 0.093] or sex [AOR = 0.91 (0.32–2.55); p = 0.855] or the season [AOR = 1.05 (0.56–1.98); p = 0.880]. The mean HAD of those concurrently wasted and stunted did not significantly differ from that of those stunted but not wasted [mean (95%CI): -9.7 (-10.5; 9.0) cm for those concurrently wasted and stunted versus -10.3 (-10.7; -10.0) cm: Δ (95%CI) = +0.6(-0.1; 1.4) cm; p = 0.098].

**Prediction of linear growth by wasting indicators.** The longitudinal data analysis revealed that both MUAC and WHZ were associated with HAZ and HAD, though there was a significant interaction between the two wasting indicators (Table 3). As shown, there was a

**Table 3. Association between the wasting and linear growth indicators among children included in the pooling analysis.** (N = 1393).

| Variable | Bivariate analysis[1] | | | Multivariate analysis[1] | | |
|---|---|---|---|---|---|---|
| | Coef[2] | (95%CI[3]) | P | Coef[2] | (95%CI[3]) | p |
| *Height-for-Age Z-score (HAZ)* | | | | | | |
| MUAC[4] (cm) | 0.18 | (0.13; 0.24) | <0.001 | 0.30 | (0.25; 0.35) | <0.001 |
| WHZ[5] | -0.24 | (-0.28; -0.19) | <0.001 | 0.46 | (0.09; 0.82) | <0.001 |
| MUACXWHZ | | | | -0.05 | (-0.08; -0.02) | <0.001 |
| HAZ at recruitment | 0.63 | (0.59; 0.67) | <0.001 | 0.57 | (0.53; 0.61) | <0.001 |
| Age (months) | -0.00 | (-0.00; 0.00) | 0.714 | -0.01 | (-0.01; -0.00) | <0.001 |
| Sex: Boys/Girls | -0.03 | (-0.27; 0.21) | 0.810 | 0.07 | (-0.06; -0.20) | 0.278 |
| Season: Lean/Post-harvest | -0.08 | (-0.17; 0.01) | 0.076 | -0.03 | (-0.11; 0.05) | 0.456 |
| Constant | | | | -5.14 | (-5.94; -4.35) | <0.001 |
| *Height-for-Age Difference (HAD)* | | | | | | |
| MUAC (cm) | -0.13 | (-0.31; 0.05) | 0.169 | 0.79 | (0.73; 0.95) | <0.001 |
| WHZ | -0.25 | (-0.41; -0.09) | 0.002 | 2.26 | (1.12; 3.41) | <0.001 |
| MUAC X WHZ | | | | -0.21 | (-0.29; -0.13) | <0.001 |
| HAD at recruitment | 0.77 | (0.72; 0.824) | <0.001 | 0.66 | (0.62; o.72) | <0.001 |
| Age (months) | -0.09 | (-0.10; -0.08) | <0.001 | -0.11 | (-0.12; -0.10) | <0.001 |
| Sex: Boys/Girls | -0.57 | (-1.45; 0.30) | 0.199 | 0.32 | (-0.15; 0.79) | 0.182 |
| Season: | -0.29 | (-0.60; 0.02) | 0.072 | -0.16 | (-0.41; 0.09) | 0.515 |
| Constant | | | | -12.01 | (-14.51; -9.51) | <0.001 |

[1]Computed using multilevel mixed effects linear regression modelling to adjust for clustering at individual and sub-district levels;

[2]Coef = coefficient;

[3]CI = confidence Interval;

[4]MUAC = Mid-Upper Arm circumference;

[5]WHZ = Weight-for Height Z-score (2006 World Health Organization growth standards).

positive relationship between wasting indicators (MUAC and WHZ) and linear growth indicators (HAZ and HAD) after adjustment for potential confounding variables (Table 3).

## Longitudinal analysis

For the 401 children evaluated, the median (IQR) of the duration between enrollment and the last assessment was 25 (16–39) months, with the extremes being 3 and 57 months. The median age shifted from 17 (7–32) months at recruitment to 52 (40–57) months at the last assessment. S1 Fig details age distribution at recruitment and at the last assessment.

**Evolution of linear growth.**   There was no significant change in mean HAZ between recruitment and the last anthropometric assessment [Δ (95%CI) = -0.01 (-0.13; 0.11); p = 0.850], despite that a positive trend in HAZ was encountered in 47.0% (187/398). Of those children who had a positive change in HAZ, 46% (86/187) had an increase in HAZ ≥ 0.67 Z-score units, indicating an accelerated linear growth. Change of stunting category was rare (Fig 4). Overall, stunting reversal was observed in only 25.6% (53/207) of children stunted at recruitment.

The assessment of linear growth between recruitment and last assessment by the mean of HAD showed that on average the length/height deficit significantly widened [Δ (95%CI) = -3.74 (-4.28; -3.21) cm; p<0.001] but catch-up trend was observed in 19.6% (78/398) with 74.4 (58/78) and 96.1 (75/78) % of the catch up episodes occurring among children not stunted and not wasted at recruitment, respectively. Those children with positive HAD trend reduced their length/height deficit by over 2 cm from a mean (SD) of -8.80 (4.51) to a mean (SD) of -6.50 (5.01) cm [Δ (95%CI) = 3.74 (-4.28; -3.21); p<0.001].

**Mediators of positive height-for-age difference.**   The results of the mediation analyses are presented in Figs 5 and 6, and Table 4. Fig 5 presents the specified l basic model examining the relationship between the two indicators of wasting and the variable representing the catch up of linear growth. It gives three important messages. First, there was full mediation of the effect of WHZ at recruitment through change in WHZ during the follow up period (Table 4). WHZ had no significant direct effect on the occurrence of a length/height catch-up as

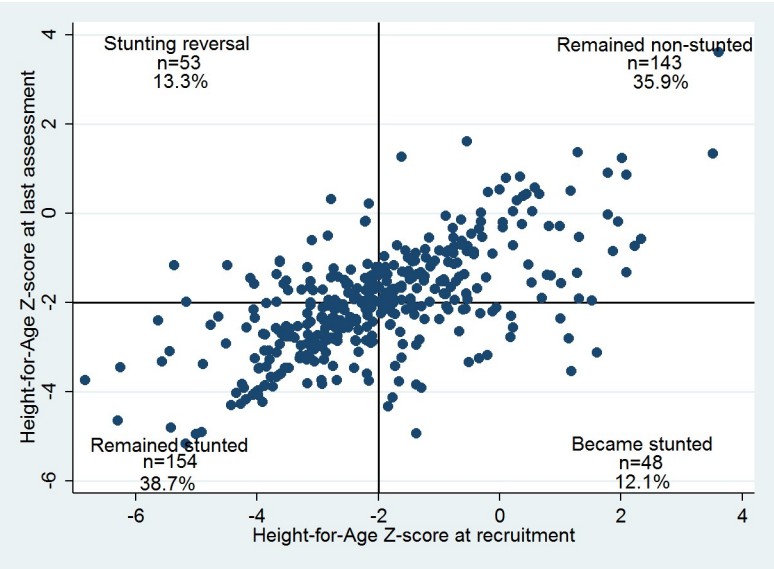

**Fig 4. Quadrant chart of change in length/height-for-age and prevalence of stunting during study participation.**

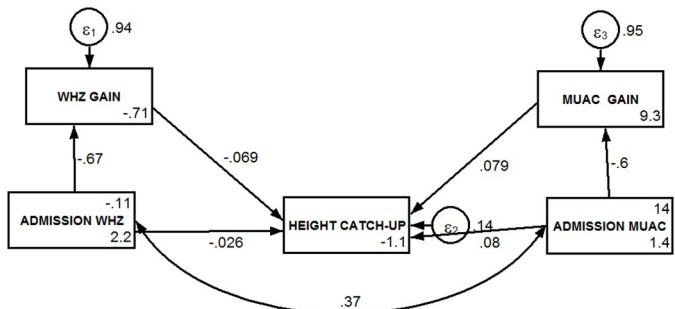

**Fig 5. Mediation analysis diagram of the association between wasting indicators and linear catch up growth.**
WHZ = Weight-for-Height Z-score, MUAC = Mid-Upper Arm Circumference; Height catch-up = Positive (>0 cm)
Height-for-Age Difference (HAD) change over the follow up period.

diagnosed by a positive HAD gain (p = 0.225), but as shown in Table 4, there was a significant
direct relationship with the mediator). Second, MUAC at recruitment had a significant
(p<0.001) positive direct effect on linear growth catch-up during the follow up period. Third,
the positive effect of MUAC at recruitment on linear growth catch-up was not significantly
influenced by the tested mediator, namely MUAC change during follow up period (Table 4).

Including the adjustment covariates, age at recruitment and at last follow up and HAD at
recruitment into the structural model revealed a significant negative direct effect of both WHZ
at recruitment (p = 0.002) and WHZ gain over the follow-up period (p<0.001) on length/
height catch-up as measured by HAD gain (Fig 6). In contrast, both MUAC at recruitment
and MUAC gain over the follow-up period retained a significant (p<0.001) direct positive
effect on HAD gain (Fig 6).

As shown in Table 4, the decomposition of the effects of all variables, even in this structural
model, is given. MUAC at recruitment had the highest positive direct effect on HAD gain, fol-
lowed by MUAC change. WHZ at recruitment and WHZ gain by the last visit had an inverse

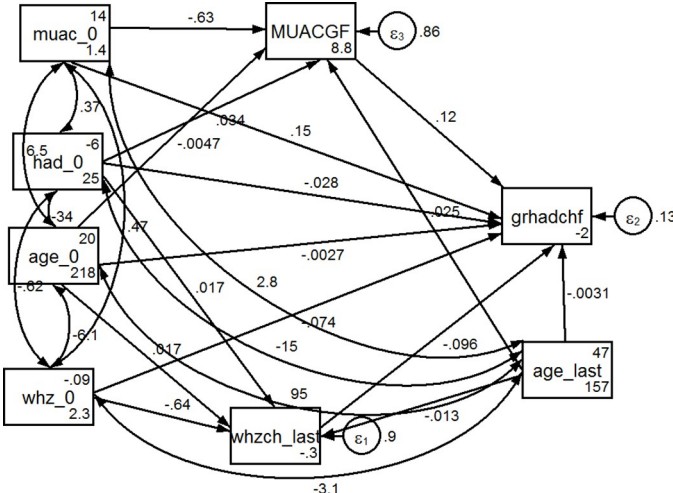

**Fig 6. Mediation analysis diagram of the association between wasting indicators and linear catch-up growth
adjusting for age, baseline height-for-age difference at recruitment, and follow up time.** WHZ = Weight-for-
Height Z-score, MUAC = Mid-Upper Arm Circumference; HAD = Height-for-Age Difference; Height catch-
up = Positive (>0 cm) Height-for-Age Difference change over the follow-up period.

**Table 4. Direct, indirect, and total effects of the path model variables in Figs 5 and 6 estimated by linear regression using the maximum likelihood approach.**

| Variable | Type of variable | Coefficient[1] | (95% CI[2]) | p-value |
|---|---|---|---|---|
| **Model of Fig 5** | | | | |
| **Direct effect** | | | | |
| WHZ[4] at recruitment | Independent | -0.026 | (-0.067; 0.016) | 0.225 |
| WHZ gain | Mediator | -0.069 | (-0.112; -0.02) | 0.002 |
| MUAC[3] at recruitment | Independent | 0.080 | (0.037; 0.123) | <0.001 |
| MUAC change | Mediator | 0.079 | (0.037; 0.120) | <0.001 |
| **Indirect effect** | | | | |
| WHZ[4] at recruitment | Independent | 0.046 | (0.017; 0.076) | 0.002 |
| MUAC[3] at recruitment | Independent | -0.047 | (-0.072; -0.022) | <0.001 |
| **Total effect** | | | | |
| WHZ[4] at recruitment | Independent | 0.021 | (-0.006; 0.048) | 0.127 |
| WHZ gain | Mediator | -0.069 | (-0.112; -0.02) | 0.002 |
| MUAC[3] at recruitment | Independent | 0.0.33 | (-0.00; 0.066) | 0.052 |
| MUAC change | Mediator | 0.079 | (0.037; 0.120) | <0.001 |
| **Model of Fig 6** | | | | |
| **Direct effect** | | | | |
| MUAC[3] at recruitment | Independent | 0.146 | (0.092; 0.199) | <0.001 |
| MUAC gain | Mediator | 0.121 | (0.077; 0.165) | <0.001 |
| WHZ[4] at recruitment | Independent | -0.074 | (-0.120;-0.027) | 0.002 |
| WHZ gain | Mediator | -0.096 | (-0.141; -0.052) | <0.001 |
| HAD[5] at recruitment | Covariate | -0.028 | (-0.037; -0.018) | <0.001 |
| Age at recruitment | Covariate | -0.003 | (-0.006; 0.001) | 0.161 |
| Age at last follow up | Covariate | -0.003 | (-0.007; 0.001) | 0.096 |
| **Indirect effect** | | | | |
| MUAC at recruitment | Independent | -0.076 | (-0.106;-0.047) | <0.001 |
| WHZ at recruitment | Independent | 0.062 | (0.033; 0.091) | <0.001 |
| HAD at recruitment | Covariate | 0.002 | (-0.001; 0.006) | 0.169 |
| Age at recruitment | Covariate | -0.002 | (-0.004; -0.001) | 0.007 |
| Age at last follow up | Covariate | 0.004 | (0.002; 0.006) | <0.001 |
| **Total effect** | | | | |
| MUAC at recruitment | Independent | 0.069 | (0.028; 0.111) | 0.001 |
| MUAC change | Mediator | 0.121 | (0.077; 0.165) | <0.001 |
| WHZ at recruitment | Independent | -0.012 | (-0.044; 0.019) | 0.445 |
| WHZ gain | Mediator | -0.096 | (-0.141; -0.052) | <0.001 |
| HAD at recruitment | Covariate | -0.025 | (-0.035; -0.015) | <0.001 |
| Age at recruitment | Covariate | —0.005 | (-0.009; -0.001) | 0.020 |
| Age at last follow up | Covariate | 0.001 | (-0.002; 0.005) | 0.495 |

[1]From Structural Equation modelling that uses linear regression modelling, the coefficients of the paths and the product method for calculating the indirect effects;

[2]CI = Confidence Interval;

[3]MUAC = Mid-Upper Arm Circumference;

[4]WHZ = Weight-For-Height Z-score (based on the 2006 WHO Growth standards);

[5]HAD = Height-for-Age Difference (Based on 2006 WHO growth standards).

relationship with linear growth catch-up in this model in which we included selected adjustment covariates.

## Discussion

The objective of this study was to examine the overlap between stunting and wasting among children of Timor-Leste and to check if their burden varies with age. Additionally, the aim was to confirm whether linear growth catch-up was demonstrated during study participation and determine the effect of wasting level at recruitment on that catch-up while checking if some of the effect is mediated through the change of wasting indicators during the follow up period.

Undernourishment early in life has profound and irreversible effects on linear growth, developmental epigenetics, and brain and neurocognitive development [58]. In our sample, stunting (54.6%) and wasting (9.5%) forms of undernutrition were well above the threshold for public health concern among children below five years of age of the surveyed rural sub-districts of Timor-Leste [13, 59]. As in many countries of the region, the prevalence of both conditions is high among children below 6 months of age, and the group level prevalence remained high or increased during the childhood period, suggesting that both prenatal and postnatal factors are fueling the burden of these conditions in the studied area [29, 33, 60–62]. The peaks of wasting and stunting were observed in the 12- to 23-month age group. The linear growth faltering process continues until the age of 3.5 years without a catch-up trend thereafter in terms of HAD positive trend. Although the HAZ mean showed an upward trend after 24 months of age, stunting reversal was observed in only a quarter of children with stunting at recruitment.

The burden of the two forms of undernutrition was within the range reported in other countries in South Asia and Southeast Asia. However, some countries have reported much higher prevalences [33, 62–65]. However, the very long period of data collection from 2009 to 2020 and the overrepresentation of older children in the analyzed pooled data call for cautiousness when comparing the observed prevalence with that of these cited references. Nonetheless, stunting and wasting were important public health problems in studied areas. Also, the most recent national surveys have reported a prevalence of 47.1% for stunting and 8.6% for wasting, which is close to what we observed, suggesting that our findings may be representative of many rural areas of Timor-Leste [13]. Thus, we believe our findings can be used to inform future nutrition and food security policies and program designs for the studied municipalities and those with similar characteristics. Additionally, our findings complement those previously published by the co-authors of this paper, highlighting the fact that a livelihood relying solely on auto-subsistence gardening and the seasonal fluctuations in household resources, including in food self-sufficiency, have a negative impact on children's growth and the importance of stabilizing these resources throughout the year [15, 41]. However, further research is needed to improve the understanding of the didtint and combined impact on health status of these two forms of undernutrition among children under 5 years in Timor-Leste and determine areas of national nutrition policy that need improvement.

This study demonstrates that the concurrence of wasting and stunting is common in Timor Leste as in other South East Asia countries [33, 34]. This finding shows that in Timor Leste, too, there is a need to integrate programs addressing wasting and stunting, as nutrition experts across the world have suggested [28, 29, 32, 66, 67]. Indeed, by showing that half of wasted children were also stunted, our findings support the integration of wasting and stunting programming and, at the minimum, the establishment of an integrated screening protocol and a bidirectional referral mechanism, allowing children diagnosed with wasting to also benefit from available package addressing stunting and vice versa [28]. This is particularly important

because community-based longitudinal studies have consistently shown that children with multiple anthropometric deficits have up to a twelve-fold increase in risk of near-term death than a child without anthropometric deficit [68–71]. Also, in terms of correcting the deficit, emerging evidence suggests that correction of wasting has a positive effect on linear growth, as shown by the temporal relationship between ponderal gain and linear growth. Several recent studies have found that stunted children respond well to treatment aiming at reversing wasting [29, 30, 56, 57, 67, 72–76].

The relationship between age and stunting prevalence observed in our study confirms what has been previously reported by co-authors of this paper and elsewhere [14, 58, 77]. As already discussed above, our results showing a high prevalence of wasting and stunting among under 6-month-old infants suggest that it is important to invest in prenatal preventive interventions. In addition, the limited decrease in the prevalence in children of older age groups highlights a limited effect of current postnatal strategies and interventions on tackling childhood undernutrition in these municipalities and probably in the entire Timor-Leste. This calls for the development and testing of new intervention approaches. In terms of wasting, the focus should be on identifying the best way of preventing its occurrence and improving the coverage of existing effective curative interventions such as the integrated management of wasting programs [4, 78, 79]. Regarding stunting, research teams are actively evaluating different types of interventions, but to date, evidence-based recommendations cannot be made for preventive and curative interventions [58, 80, 81].

In accordance with the findings of several previous studies that investigated the temporality between length/height and weight gain, we also found an inverse relationship between WHZ and HAD gain, suggesting that children who accelerate linear growth did so at the expense of body weight [29, 30, 56, 57, 67, 72–74]. The most frequently proposed hypotheses to explain this growth pattern are: 1) a combination of a preferential mobilization of body reserves and an antagonism between the molecular mechanisms promoting weight gain and that stimulating linear growth catch-up and 2) failure to increase dietary intake during the spurts of linear growth [82–84]. The first hypothesis can be backed by studies that showed that body weight replenishment after growth stagnation during the lean season or after an episode of wasting is needed for linear growth to resume [57, 67, 72, 73]. For instance, the antagonism between weight gain and height gain processes may ensure that sufficient ponderal gain and ample body reserves in essential nutrients precede linear growth spurts, creating the frequently observed lag between weight gain and length/height gain [57, 67, 72, 84]. This antagonism can also explain the occurrence of wasting during the episode of rapid linear growth as the weight gain process is inhibited at the very same time of excess body fat and body muscles are used for height growth. However, this hypothesis cannot fully explain our findings. Indeed, if the influence of mutual inhibition of molecular mechanisms promoting weight gain and that stimulating linear growth may have played a role, it is unlikely that the negative direct relationship observed between WHZ at recruitment and positive change in HAD was due to the preferential mobilization of the already stored nutrients [82, 83]. Indeed, we can assume that higher WHZ at recruitment will mean higher body reserves of required nutrients and vice versa. Thus, the results should have been that MUAC also would have had that same inverse relationship with the increase in HAD. The second hypothesis suggesting that failure to adjust macronutrients and micronutrients dietary intake during a period of increased energy and nutrient requirements may be the main determinant of poor weight gain is supported by some clinical experiences and epidemiological community-based studies [84, 85]. For instance, studies on the treatment of short stature conducted in high-income countries showed that an increase in dietary intake was necessary to avoid weight loss during treatment, such as using growth hormones [86, 87]. Also, the frequently observed relationship between stunting levels and

households' socio-demographic characteristics supports this hypothesis. Finally, the fact that this temporal pattern in weight and height gain was not observed among children used to develop the current WHO growth references suggests that this is not the physiological norm but a pattern influenced by child living conditions [85, 88]. However, if this mechanism contributes to the development of wasting, it cannot explain the direct relationship between MUAC and HAD positive change and the lag between weight gain and height gain. All the above indicate that more research is needed to improve our understanding of the cause of the development of wasting during catch-up linear growth.

Recently, a new hypothesis has been suggested to explain the lag between weight gain and height gain during early childhood for children living in settings with growth-limiting conditions [84]. Based on the data from 5039 Burkinabè children enrolled at 6 months in a prospective cohort study that implemented monthly anthropometric (weight, length) measurements during consecutive 21 months, the authors concluded that the same growth-limiting or growth-promoting conditions affect both the ponderal and linear growth processes concurrently and that weight and length increase and decrease in parallel with a lag of 1 month due to the fact that linear growth is a slower process than ponderal growth [84]. This conclusion was adopted based on the overall trend, but detailed analyses revealed that faster ponderal growth was associated with faster concurrent and subsequent linear growth, while faster linear growth was associated with slower future weight gain [84]. Although the results of this study strongly support this conclusion, the fact that all these Burkinabe children were receiving specially formulated supplementary food aimed at preventing stunting and wasting limits the interpretation. It is likely that the 18 months of consumption of the energy and nutrient-enriched food modified the growth pattern [84]. Nonetheless, the finding provides evidence that preventing or reversing wasting has the potential to improve linear growth.

Several prospective cohort studies have demonstrated that for children growing in a growth-limiting environment, the decline in HAZ overtime is more important among children who were less stunted and probably also less wasted at enrolment than in those having the worse anthropometric parameters at the beginning of the follow-up [29, 84, 89]. Thus, it cannot be excluded that the inverse relationship between WHZ and HAZ observed in our study over the median interval of 25 months is mostly reflecting the slight recuperation of children with the worse linear growth parameters at enrolment and the rapid decline of these parameters in those who were better-off.

Our finding showing that MUAC and WHZ influenced differently the likelihood of experiencing linear growth catch up as demonstrated by a HAD positive change (our principal outcome) is in accordance with what was observed in a study investigating the relationship between wasting anthropometric parameters and linear growth among Cambodian children that showed a lag of 3 to 4 months between WHZ and HAD increase but no lag between MUAC change and HAD gain [29]. It also echoes the finding of one study conducted in Ethiopia that compared the use of MUAC and WHZ for monitoring the response to therapeutic feeding and found that WHZ and MUAC do not always increase in parallel during nutrition recovery from wasting [90]. These findings suggest that subcutaneous fat is less mobilized during spurts of linear growth than during the early phase of starvation, hence a preferential use of visceral fat to support the linear growth process [91]. Indeed, it has been shown that visceral and subcutaneous fat tissues' response to starvation does not occur simultaneously and that in some individuals, visceral fat tissues are first used as an energy source to compensate for the restriction in energy supply from the diet [91].

The contrasting relationship between WHZ and HAD and MUAC and HAD definition, above also suggests that there are biological mechanisms tightly controlling arm growth in children aged 6 to 59 months, which explains why MUAC is not a useful tool for monitoring

response to treatment in wasted children who did not meet the MUAC wasting definition but it is useful for children who had low MUAC at the start of treatment [54, 90]. Another possible explanation for the discrepancy between weight and MUAC changes is a tight correlation between MUAC and lean mass, including skeletal muscle, and the level of leptin synthesis and release [83, 92]. The good correlation between MUAC and body muscle has led to the development of different arm and height ratios currently used for the diagnosis of obesity and monitoring the effect of its treatment [93, 94].

Cliffer et al. have suggested that the investigation of dependency of weight and height growth trends during early childhood in low and middle-income countries using attained weight, height, and their respective indices, as used in many previously published studies, leads to a misleading conclusion [29, 57, 67, 73, 84, 85, 95, 96]. They propose to use velocities and a short interval between follow-ups. It has been recently demonstrated that HAD is the best metric for measuring and describing catch-up growth [96–98]. In this study, we used positive HAD gain to assess and define linear growth catch-up at an individual level. Thus, it can be considered that we used the best available metric. Interestingly, our findings are in accordance with the findings of the studies that used other metrics that strengthen the body of the evidence, especially regarding the temporal relationship between weight and length/height velocities.

Despite MUAC being used for a decade as an indicator of nutrition status, especially as an alternative to WHZ for diagnosing wasting, the relationship between MUAC and linear growth has received less attention than that of WHZ and linear growth [48, 99–101]. However, similar to our findings, the data analysis of a longitudinal study of Cambodian children showed a distinct relationship of WHZ and MUAC with accelerated linear growth [29]. In accordance with our results demonstrating significant direct positive relationships of MUAC at recruitment and of MUAC change during follow up to positive HAD gain, the Cambodian study showed a positive correlation of MUAC change during the follow-up period and accelerated linear growth and a lagged positive correlation with this outcome [29]. These results support proposing MUAC as a tool to detect the effect of intervention on linear growth where direct height measurement is not feasible. However, more studies are needed to confirm this relationship, enhance understanding of the molecular mechanisms, and assess the usefulness of MUAC in predicting or as a proxy measure of linear growth.

In contradiction with the frequently reported association between lean season and the increase in wasting prevalence as measured by the WHZ indicator, we found that it was not the case in our study [57, 72, 102–105]. Few other studies have reported similar observations, and several explanations have been proposed, including high food insecurity levels all year round, less reliance on auto-subsistence agriculture with more involvement in the production of cash crops, preventive safety net interventions such as cash transfer, targeted general food ration and blanket supplementary feeding programs [106–108]. Some of these explanations can apply to our study setting. For instance, it was reported that less than half of the population of Natarbora rely solely on their own agricultural yield [41]. However, the discrepancy between the seasonal effect on WHZ-based prevalence and MUAC-based prevalence suggests that losing body mass affected body weight and body circumferences differently. This may occur if the loss of lean muscles is masked by a body composition change, leading to relative overhydration if the children preferentially lose body fat rather than lean mass [109–111]. Thus, our results may also indicate that MUAC is more sensitive to short-term and small tissue loss than WHZ.

Ensuring a harmonious ponderal and linear growth to prevent wasting and stunting under-five children is a global public health priority. Unfortunately, available interventions designed based on some of the hypotheses mentioned above have yielded contrasting results [80, 112,

113]. Thus, more research is needed to improve the understanding of determinants of weight and length/height during childhood growth, a prerequisite for designing effective wasting and stunting interventions [73, 85, 96, 114, 115]. Globally, there is a consensus based on experts' opinions that countries need to implement at-scale multi-sector interventions combining nutrition-sensitive and nutrition-specific programs to significantly reduce the burden of childhood stunting [12, 116–118]. Our findings of the important contribution of intra-uterine growth restriction to stunting burden, that of low proportion of children reversing stunting combined with previously published evidence on height trajectories of children of the study setting showing early growth faltering and a height curve with the median tracking below the 5th percentile of the WHO reference curve throughout childhood and adolescence advocate for the systematic inclusion in the multi-sector package of interventions targeting women of reproductive age as these interventions have the potential of improving intra-uterine and early postnatal growth [39, 80, 119]. For example, maternal malnutrition may negatively affect breast milk production; hence, preventing or reversing it may eliminate the negative effect of insufficient breast milk production [120, 121]. Also, a study in India demonstrated that maternal characteristics during pregnancy significantly influenced the incidence of common infectious diseases during the first years of life of offspring [122].

Our findings have to be interpreted taking into account the study's strengths and limitations. The first main strength is using mixed effects regression analysis for repeated measurements to estimate prevalences and bivariate associations. The second strength of the study is the use of HAD gain as an indicator of linear growth catch up as new evidence has shown that it is more meaningful to use it than to use HAZ [97] as HAZ may appear to stabilize or increase even in cases where height deficits continue to accumulate. The third strength is the use of mediation analysis to determine the relationship between the predictors and HAD gain, as mediation analysis allows distinct estimation of direct effect and effect mediated by a set of mediators/confounders [123, 124]. The study's fourth strength was the exhaustiveness of the sampling under-five children of the targeted communities [41]. The main weaknesses were the wide range of follow up duration of 3 to 57 months due to the design of the original study of an open cohort with new children enrolled at each data collection round rather than just following the same cohort throughout the study period and for the fact that some potential determinants including child feeding history and practices, socio-economic and household's characteristics were not included in the models. We are unsure how these factors affected the interpretation and conclusions of our study. However, the effect of the design of the original study may have affected the estimation of the burden of both wasting and stunting, but we believe that this design issue has not affected the association between the wasting parameters and the linear growth catch-up indicator we used. In terms of the duration variation, we controlled for this factor in our structural models; thus, we think this factor did not introduce a bias. Also reassuring is the fact that our analysis yielded conclusions similar to those found in studies standardized follow up intervals and periods [67, 83, 85].

## Conclusion

In conclusion, child undernutrition, including stunting and wasting, remains widespread among rural children under five years of age in Timor-Leste, with indications that prenatal growth restriction is contributing a large share of the burden. New approaches for tackling stunting and wasting in Timor Leste are needed. Our findings provide new evidence that stunting and wasting should no longer be regarded as unrelated nutritional diseases and advocate for the change in the way the two are currently described and addressed and for testing the effect of the integration of stunting and wasting programming on their respective burden,

especially in countries where the priority is given to only one of these forms of undernutrition. Also underscoring the need for this integration is that in Timor-Leste and many affected countries, the prevalences of both conditions remain above the threshold for public health concern and because of the proportion of the concurrence of the two forms of undernutrition. More research is needed to understand the physiopathology of wasting and wasting and the relationship between ponderal and linear growth. An improvement in the understanding of the determinants and pathophysiology of these health conditions is urgently needed and should be included in the Timor-Leste and global list of research priorities.

## Supporting information

**S1 Fig. Kernel density of children age at recruitment and at last assessment.**
(TIF)

## Acknowledgments

The authors would like to thank Roland Kupka and Paul Binns for their insightful comments that improved the manuscript. We also thank the University of Western Australia research team for sharing the dataset with us. Thanks also to the UNICEF Timor Leste, UNICEF East Asia and the Pacific Regional office (EAPRO), and Action Against Hunger teams for the administrative and technical support provided throughout the study implementation.

**Disclaimer**: Mueni Mutunga and Faraja Chiwile are UNICEF staff members. The opinions and statements in this article are those of the authors and may not reflect official UNICEF policies.

## Author Contributions

**Conceptualization:** Paluku Bahwere, Debra S. Judge, Phoebe Spencer, Mueni Mutunga.

**Data curation:** Paluku Bahwere, Debra S. Judge, Phoebe Spencer.

**Formal analysis:** Paluku Bahwere.

**Funding acquisition:** Debra S. Judge, Phoebe Spencer.

**Investigation:** Debra S. Judge, Phoebe Spencer.

**Methodology:** Paluku Bahwere.

**Project administration:** Faraja Chiwile, Mueni Mutunga.

**Supervision:** Debra S. Judge, Phoebe Spencer.

**Validation:** Debra S. Judge, Phoebe Spencer, Faraja Chiwile, Mueni Mutunga.

**Writing – original draft:** Paluku Bahwere.

**Writing – review & editing:** Debra S. Judge, Phoebe Spencer, Faraja Chiwile, Mueni Mutunga.

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
