## [Decision Letter · Decision Letter 0]

23 Jan 2024

PONE-D-23-40750Examining the burden and relationship between stunting and wasting among Timor-Leste under five rural childrenPLOS ONE

Dear Dr. Bahwere,

Thank you for submitting your manuscript to PLOS ONE. After careful consideration, we feel that it has merit but does not fully meet PLOS ONE’s publication criteria as it currently stands. Therefore, we invite you to submit a revised version of the manuscript that addresses the points raised during the review process.

We look forward to receiving your revised manuscript.

Kind regards,

Dhruba Shrestha, MD

Academic Editor

PLOS ONE

“The authors would like to thank Roland Kupka and Paul Binns for their insightful comments that improved the manuscript. We also thank the University of Western Australia research team for sharing the dataset with us. Thanks also to the UNICEF Timor Leste, UNICEF East Asia and the Pacific Regional office (EAPRO) and Action Against Hunger teams for the administrative and technical support provided throughout the study implementation. The original study was funded by the Australian Research Council (Grant DP 120101588), the School of Anatomy, Physiology, and Human Biology at the University of Western Australia, and the Timor-Leste  UNICEF  office. The funders played no role in the design of this secondary data analysis study and the interpretation of the results”

“DSJ and PS received funding for the data collection for the original study from the Australian Research Council. the School of Anatomy, Physiology, and Human Biology at the University of Western Australia and UNICEF Timor-Leste.”

Reviewers' comments:

Reviewer's Responses to Questions

**Comments to the Author**

1. Is the manuscript technically sound, and do the data support the conclusions?

Reviewer #1: Yes

Reviewer #2: Yes

2. Has the statistical analysis been performed appropriately and rigorously? 

Reviewer #1: Yes

Reviewer #2: Yes

3. Have the authors made all data underlying the findings in their manuscript fully available?

Reviewer #1: Yes

Reviewer #2: Yes

4. Is the manuscript presented in an intelligible fashion and written in standard English?

Reviewer #1: Yes

Reviewer #2: Yes

5. Review Comments to the Author

Reviewer #1: Dear Authors

Thank you for this manuscript. I have minor issues to be considered.

1. As I understand you used secondary data from previous surveys, in the data collection section, you describe how the surveys were conducted. I think it is important that you describe how you conducted data collection for this paper, how did you select the surveys, who did that, where did you get the sources, were they in your possession or did you have to acquire from government institutions?

2. Adequate nutrition during infancy and early childhood is essential to ensure the growth, health, and development of children to their full potential. Stunting and wasting is influenced by nutrition. How was nutrition during infancy and childhood in the surveys controlled? In your statistical analysis to determine the relationship between stunting and wasting, did you use feeding in the regression analysis?

3. Since this is not a new subject, state clearly the new knowledge you are bringing to literature.

Thank you.

Reviewer #2: Dear Authors,

Thank you for the chance to me for review the manuscript entitled: Examining the burden and relationship between stunting and wasting among Timor-Leste under five rural children. This manuscript is very important to solve public health problem, particularly for under-five children with stunting issues in low-middle income countries. However, before to publish this manuscript, there are any issues that should add and clarify to improve this manuscript.

1. In title the authors mentioned about rural areas, please describe some issues about fulfilling nutrition for under-five children in the rural areas, especially in low-middle income countries.

2. This research was conducted in Timor-Leste, please describe any program that government done to solve this problem. Therefore, we know the importance this study should be done.

3. In the method. Please describe more about Timor-Leste. How many public health centers on there and how they develop program to solve stunting. How many areas that included in this study and how was done to collect the data in this study.

4. The results are well-written and reflected the study.

5. Discussion. The authors said that they are missed about stunting and wasting in Timor Leste. Please describe how the healthcare providers on there to measure about height and weight. We think that measurement and recorded the assessment in public health setting is important issues. Sometimes, they have limited equipment or missed to record in growth chart, etc.... please describe more about the field observation on there and discussed.

6. Conclusion is well-written and answered the objective of study.

7. Reference. Please add some references from Indonesia and Papua Nugini to comapre and contract the findings, because this is some areas with stunting cases.

Thank you.

Kind regards,

Reviewer

6. PLOS authors have the option to publish the peer review history of their article (what does this mean?). If published, this will include your full peer review and any attached files.

Reviewer #1: No

Reviewer #2: No

---

## [Author Response · Author response to Decision Letter 0]

11 Apr 2024

We have attached a file responding to all comments and queries.

---

## [Decision Letter · Decision Letter 1]

8 Oct 2024

Examining the burden and relationship between stunting and wasting among Timor-Leste under five rural children

PONE-D-23-40750R1

Dear Dr. Paluku Bahwere,

We’re pleased to inform you that your manuscript has been judged scientifically suitable for publication and will be formally accepted for publication once it meets all outstanding technical requirements.

Kind regards,

Dhruba Shrestha, MD

Academic Editor

PLOS ONE

Additional Editor Comments (optional):

Reviewers' comments:

Reviewer's Responses to Questions

**Comments to the Author**

1. If the authors have adequately addressed your comments raised in a previous round of review and you feel that this manuscript is now acceptable for publication, you may indicate that here to bypass the “Comments to the Author” section, enter your conflict of interest statement in the “Confidential to Editor” section, and submit your "Accept" recommendation.

Reviewer #1: All comments have been addressed

Reviewer #3: All comments have been addressed

2. Is the manuscript technically sound, and do the data support the conclusions?

Reviewer #1: Yes

Reviewer #3: Yes

3. Has the statistical analysis been performed appropriately and rigorously? 

Reviewer #1: Yes

Reviewer #3: Yes

4. Have the authors made all data underlying the findings in their manuscript fully available?

Reviewer #1: Yes

Reviewer #3: Yes

5. Is the manuscript presented in an intelligible fashion and written in standard English?

Reviewer #1: Yes

Reviewer #3: Yes

6. Review Comments to the Author

Reviewer #1: Dear Authors

Thank you for considering the recommendations made. I am satisfied that you have responded to them well.

Reviewer #3: Malnutrition, mainly wasting and stunting, are two major subjects of interest globally, especially in developing countries and the burden is increasing in developed countries as well. It is a good article with details addressing all required areas of concern.

7. PLOS authors have the option to publish the peer review history of their article (what does this mean?). If published, this will include your full peer review and any attached files.

Reviewer #1: No

Reviewer #3: **Yes: **Nipun Shrestha

---

## [Editor Report · Acceptance letter]

16 Oct 2024

PONE-D-23-40750R1 

PLOS ONE

Dear Dr. Bahwere, 

I'm pleased to inform you that your manuscript has been deemed suitable for publication in PLOS ONE. Congratulations! Your manuscript is now being handed over to our production team.

Kind regards, 

on behalf of

Dr. Dhruba Shrestha 

Academic Editor

PLOS ONE